# Exploring the Biodiversity and Conservation Value of Alpine Grasslands in the Bucegi Massif, Romanian Carpathians

Claudia Bita-Nicolae [1,*] , Faruk Yildiz [2] and Ozkan Kaya [3]

1 Ecology, Taxonomy & Nature Conservation Department, Institute of Biology Bucharest, Romanian Academy, 060031 Bucharest, Romania
2 Institute of Science, Erzincan Binali Yıldırım University, 24002 Erzincan, Turkey
3 Erzincan Horticultural Research Institute, Republic of Turkey Ministry of Agriculture and Forestry, 24060 Erzincan, Turkey; kayaozkan25@hotmail.com
* Correspondence: claudia.bita@ibiol.ro

**Abstract:** The Carpathian region harbors a wide range of threatened species, making it an area of exceptional conservation value. In the Alpine belt, grasslands cover the entire region and this study aims to describe the communities in the Bucegi Massif of the Romanian Carpathians and highlight their importance for conservation. The Braun–Blanquet approach was used to record floristic data from 47 phytosociological surveys, identifying a total of 235 plant species from 40 different families, including 30 threatened species. Canonical correspondence analysis was used to analyze the data, revealing that the distribution of vegetation is mainly influenced by elevation, slope and vegetation cover. Dominant grass species in these communities include *Nardus stricta*, *Festuca violacea*, *Kobresia myosuroides*, *Festuca amethystina*, *Festuca airoides*, *Sesleria rigida*, *Festuca versicolor* and *Festuca carpatica*. The alpine and boreal siliceous grasslands of the Carpathian Mountains, identified by Natura 2000 codes 6150, 6130 and 6170, host a wide range of plant species of significant conservation value. The higher altitude grasslands, especially, have outstanding plant species richness. We argue that although the habitats have been grazed, significant parts of the area are still in good ecological condition, having many typical natural features.

**Keywords:** alpine grasslands; plant species; high biodiversity; community; phytocoenology; threatened species





## 1. Introduction

Alpine grasslands are found in mountainous regions around the world [1–7] at high altitudes where harsh environmental conditions such as low temperatures, high winds and frequent snow and ice cover limit the growth of trees and other vegetation [2–4]. Grasslands with about 50% of European endemic plant species [5–7] account for an impressive 20% of Europe's vascular plant flora [7–10], although they cover only 3% of the total land area [11–14]. Alpine grasslands are characterized by high biodiversity and endemism [15,16], and also provide important ecosystem services such as carbon sequestration, soil conservation and water regulation, but are often associated with traditional land-use practices [11,15,16] and cultural heritage [17].

At the EU level, the Birds and Habitats Directives [18] serve as policy instruments for safeguarding natural and semi-natural habitats, including grasslands and improving their conservation status. The Natura 2000 [19] network comprises sites of community interest, encompassing 198 habitat types listed in Annex I of the Habitats Directive, among them 26 pasture habitats and six grassland habitats threatened by the abandonment of pastoral-management practices. To address biodiversity threats in agricultural landscapes [17,18], legal and administrative measures have been implemented. Despite its extensive coverage—over 25,000 sites and 1 million km$^2$—the Natura 2000 network primarily focuses on habitats with high biological values, notably semi-natural grasslands [19–21]. Palearctic grasslands,

essential for diverse ecosystem functions and services [21–23], are intricately linked to their biodiversity [24–26]. Regrettably, these grassland communities and their biota encounter severe threats to their survival, including habitat degradation, invasive species, and climate change [27–29]. The Bucegi Mountains, located in the Carpathian chain [30], have a longstanding tradition of sheep grazing in alpine grasslands, that dates back to the 16th century [25,31]. However, the increasing size of sheep flocks over time has resulted in negative impacts on the area's plant diversity, vegetation structure and soil erosion [31]. Moreover, grazing has spread to adjacent forests that were clearcut in the 19th century to provide additional pasture land [30,31]. Since 1989, when the socio-economic situation changed, the intensification of sheep grazing has expanded to steep mountain slopes, which is concerning due to the presence of endemic and relic plant species in the Southern Carpathians [32].

1. Alpine grasslands have become a topic of intense discussion worldwide due to their ecological [2,3,7,10,16,22,33] and cultural significance [14,34]. The Western Carpathians and also the South-Eastern Carpathians, where our study area is located, are no exception to this trend [34,35]. The first in-depth studies of alpine grasslands in this region were carried out by Pușcariu et al. [36] and Beldie [37], focusing on the endemic flora found in the area and the conservation value of these grasslands [36]. However, a research gap remains regarding the specific ecological impact of intensified sheep grazing on these alpine grasslands in the study area. Recent studies have shown that grasslands on steep slopes possess significant biodiversity value [38,39], making it imperative to investigate the effects of intensified sheep grazing on these grasslands [31]. Understanding the ecological consequences of this grazing practice will help identify potential conservation measures and management strategies to maintain the biodiversity and cultural significance of alpine grassland in the study area [40,41]. In addition to the negative impacts of grazing, activities such as infrastructure development, tourism [10,12,34,35], and climate change-induced effects may contribute to the fragmentation and degradation of habitats in the region [7,15,37]. Thus, the primary objectives of this study are as follows: (I) Identification of alpine grassland plant communities: The study aims to identify and characterize the alpine grassland plant communities present in the study area. (II): Assessment of remaining plant species: The research seeks to assess the diversity and distribution of plant species in the study area, particularly focusing on endemic, rare, and vulnerable species, which play a crucial role in the region's ecological and conservation significance. (III): Description of priority natural habitats: The study also aims to describe and prioritize the natural alpine grassland habitats found in the Bucegi Massif. Understanding the importance of these habitats is essential for formulating effective conservation strategies and protecting the unique biodiversity of the area. (IV) Raising Awareness: The study aims to highlight the significance of endemic, rare and vulnerable plant species; the research intends to contribute to their conservation and management..

## 2. Materials and Methods

### 2.1. Area of Study

The Bucegi Massif is situated at coordinates 45.41° N and 25.45° E, and is part of the Romanian Carpathians, as shown in Figure 1 (we used the GNU Image Manipulation Program (GIMP)) [42]. Covering 47% of Romania, the Carpathian Mountains are the largest mountain range in Europe, with the Romanian Carpathians representing 55% of this area [43]. They are also known for their exceptional conservation value, with rich endemic taxa and high levels of biodiversity [44].

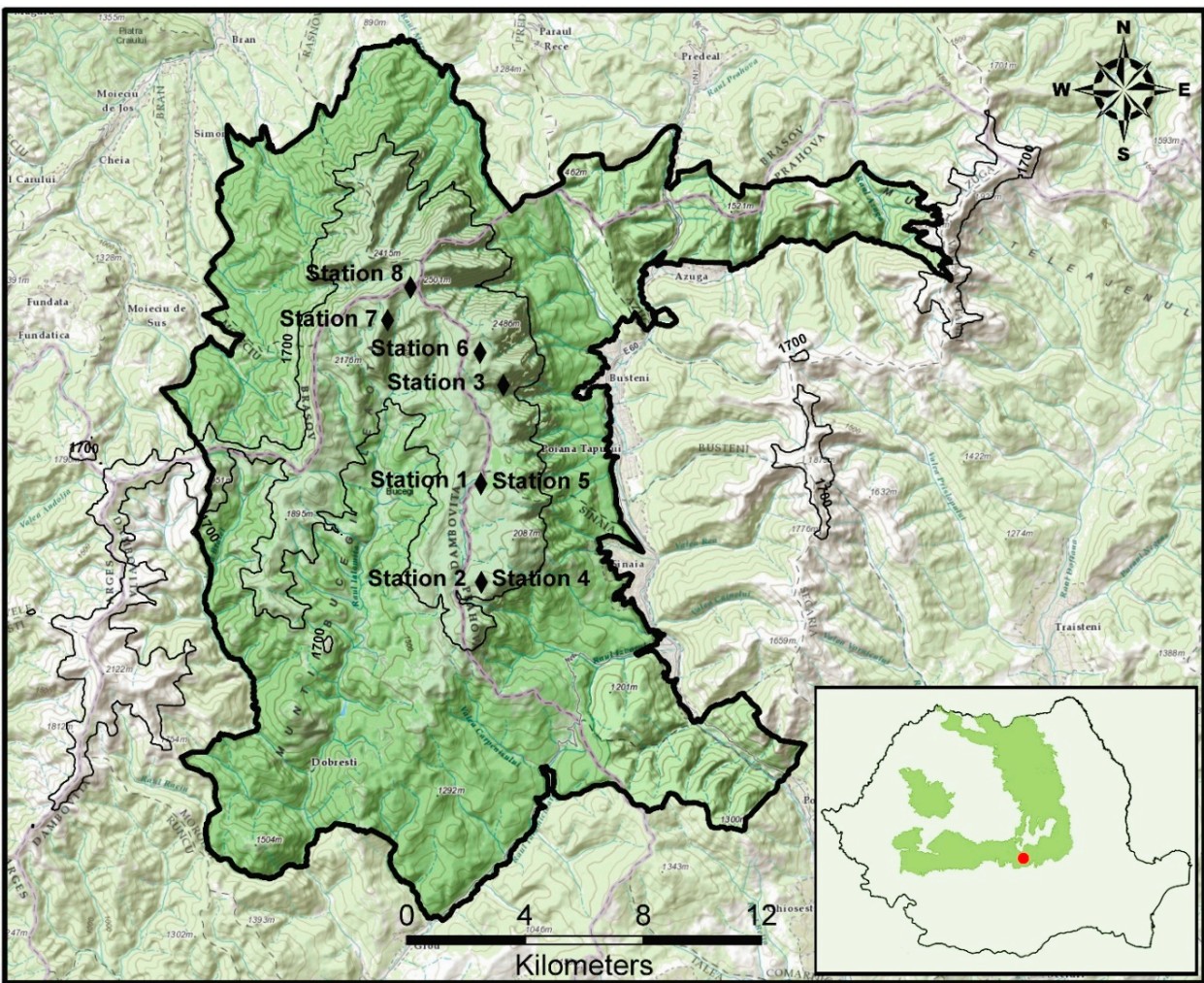

**Figure 1.** The studied territory, Bucegi Massif of the South-Eastern Carpathians.

The Bucegi Massif (Figure 1), with an altitudinal range of 798 to 2505 m and annual average rainfall ranging from 177 to 1423 mm, is a particularly interesting area for the floristic survey. The Bucegi Conglomerate, a Cretaceous formation with significant limestone content, is the predominant rock type in the mountains [45]. The location and rugged topography strongly influence the distribution and unilateral or multiple effects of general climatic or microclimatic factors [46].

As a result, the climate of the peaks is cold, with strong north-westerly winds, which drive clouds, leading to rich precipitation. The average annual temperature is about 0 °C and the average temperature of the warmest month is 10 °C [47]. The average annual rainfall at altitudes of 1400–1800 m is about 1100 mm per year [48] (Figure 2).

Given its ecological significance, the Bucegi Massif has been granted conservation status as a Natural Park and is recognized as a Natura 2000 site (RO SCI 0013 Bucegi) [49].

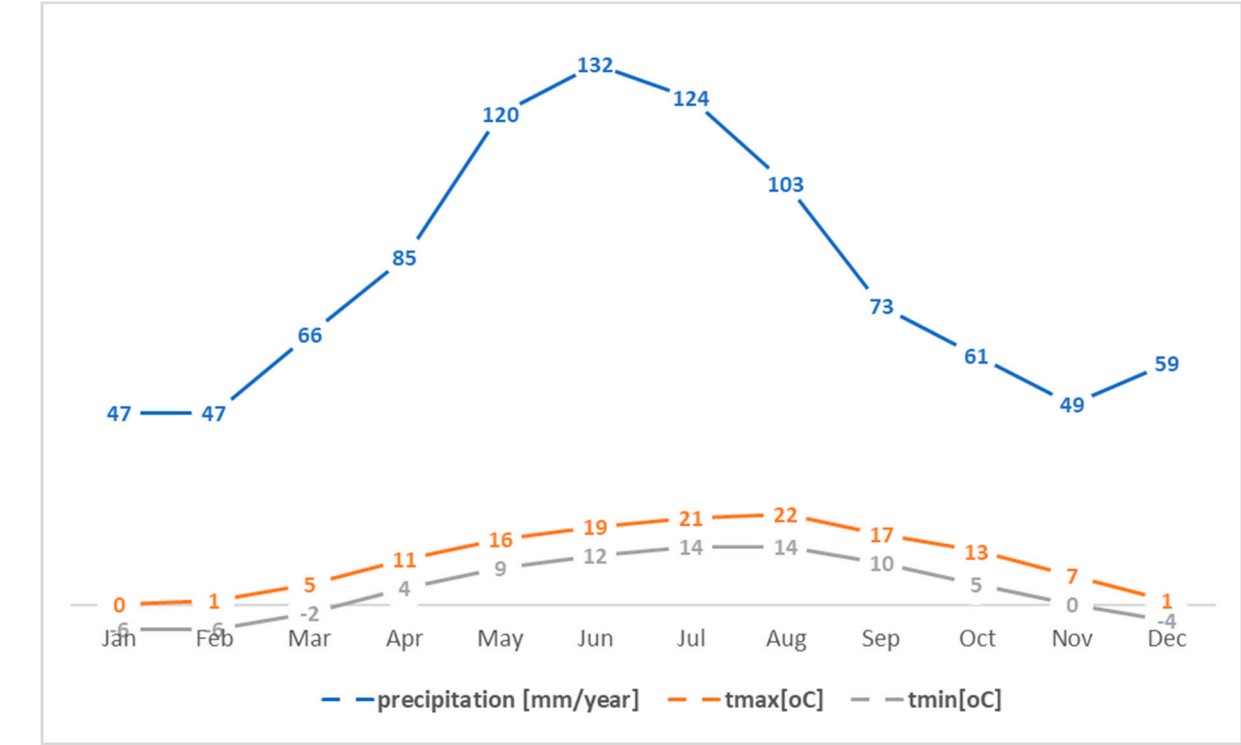

**Figure 2.** Average temperatures and precipitation in the Carpathians [48].

*2.2. Field Data Collection*

This study aimed to conduct a survey of the Bucegi Massif, South-Eastern Carpathians, during the summers of 2018 and 2020 to identify the plant communities. The survey involved collecting floristic data from 45 (Table 1) randomly selected plots, each measuring between 4 and 100 m² (Table 1). Field sites were selected subjectively. Floristic and geographical data were collected for each plot in stations 1 to 8 (Figure 1). Other parameters, such as the general topography of the area (altitude, exhibition, and inclination), were also recorded. The Braun–Blanquet approach [50] was utilized to record floristic data on each plot, with nomenclature in line with Romanian studies. Flora Illustrata of Romania [51], Flora României [52] and Flora Europaea [53] were used to identify species. The nomenclature for syntax followed the model of Mucina et al. [54].

**Table 1.** Parameters for the following: Community of *K. myosuroides*: 1–4, Community of *F. amethystina*: 5–11, Community of *S. rigida*: 12–18, Community of *F. versicolor*: 19–23, Community of *F. carpatica*: 24–28, Community of *F. airoides*: 29–34, Community of *F. violacea*: 35–40, Community of *N. stricta*: 41–46.

| No. | 1 | 2 | 3 | 4 | 5 | 6 | 7 | 8 | 9 | 10 | 11 | 12 |
|---|---|---|---|---|---|---|---|---|---|---|---|---|
| Altitude (m) | 2250 | 2250 | 2200 | 2200 | 2200 | 2100 | 1950 | 1900 | 1780 | 2000 | 2060 | 2200 |
| Exhibition | E | E | E | E | S | SE | SE | E | E | SV | E | VE |
| Inclination (degrees) | 15 | 15 | 20 | 10 | 40 | 25 | 35 | 50 | 45 | 35 | 45 | 5 |
| Vegetation cover (%) | | 35 | 45 | 20 | 75 | 55 | 55 | 65 | 75 | 80 | 55 | 25 |
| Area (m²) | 4 | 4 | 4 | 4 | 50 | 25 | 25 | 50 | 50 | 50 | 50 | 50 |
| No. | 13 | 14 | 15 | 16 | 17 | 18 | 19 | 20 | 21 | 22 | 23 | 24 |

**Table 1.** *Cont.*

| No. | 1 | 2 | 3 | 4 | 5 | 6 | 7 | 8 | 9 | 10 | 11 | 12 |
|---|---|---|---|---|---|---|---|---|---|---|---|---|
| Altitude (m) | 1900 | 2100 | 1700 | 1800 | 800 | 750 | 2000 | 2100 | 2300 | 2200 | 2100 | 2000 |
| Exhibition | E | E | S | S | V | V | SE | SE | SE | SE | E | E |
| Inclination (degrees) | 60 | 30 | 40 | 25 | 75 | 80 | 25 | 25 | 45 | 30 | 20 | 45 |
| Vegetation cover (%) | 75 | 70 | 100 | 90 | 20 | 40 | 60 | 30 | 30 | 60 | 60 | 70 |
| Area (m$^2$) | 25 | 50 | 100 | 100 | 4 | 4 | 4 | 4 | 4 | 4 | 4 | 16 |
| No. | 25 | 26 | 27 | 28 | 29 | 30 | 31 | 32 | 33 | 34 | 35 | 36 |
| Altitude (m) | 1900 | 1820 | 1750 | 1800 | 1800 | 1800 | 1730 | 1800 | 1650 | 1700 | 1500 | 1550 |
| Exhibition | E | NE | NE | NE | E | E | E | V | V | S-V | E | E |
| Inclination (degrees) | 40 | 35 | 45 | 30 | 10 | 15 | 5 | 10 | 25 | 20 | 75 | 75 |
| Vegetation cover (%) | 40 | 30 | 50 | 40 | 75 | 65 | 60 | 80 | 100 | 80 | 30 | 35 |
| Area (m$^2$) | 16 | 16 | 50 | 50 | 100 | 100 | 100 | 100 | 100 | 100 | 100 | 100 |
| No. | 37 | 38 | 39 | 40 | 41 | 42 | 43 | 44 | 45 | | | |
| Altitude (m) | 1600 | 1500 | 1700 | 1600 | 1900 | 1850 | 1910 | 2000 | 1880 | | | |
| Exhibition | E | S | S | V | E | E | E | E | V | | | |
| Inclination (degrees) | 90 | 85 | 80 | 75 | 20 | 15 | 10 | 20 | 15 | | | |
| Vegetation cover (%) | 30 | 20 | 15 | 20 | 75 | 75 | 90 | 80 | 80 | | | |
| Area (m$^2$) | 50 | 100 | 50 | 100 | 100 | 100 | 100 | 100 | 100 | | | |

*2.3. Data Analysis*

We used the Shannon diversity index (H') to analyze the distribution of species abundance across the entire dataset [55], taking into account both the number of individuals and the number of species present [56]. The evenness index was also used to assess whether the distribution of species abundance in the study area was uniform or skewed towards certain dominant species.

To reduce noise in the data, species that were present in each relevé with a count of five or less (N ≤ 5) were removed [57]. Additionally, both the species response and explanatory variables were logarithmically transformed to allow for the comparison of response variables measured on different scales.

We assessed the distribution of the data using the Shapiro–Wilk test and then performed a canonical correspondence analysis (CCA) to explain the variation in floristic composition based on environmental variables [58]. By conducting the Shapiro–Wilk test, we aimed to meet the assumptions necessary for the selected statistical methods. This allows us to make accurate inferences and draw meaningful conclusions from the data. We used the Monte Carlo permutation test with 499 permutations to determine the significant influence of environmental variables on species composition, with a significance level of $p < 0.01$. The statistical analysis of the dataset was conducted using the CANOCO version 4.5 software package [58]. To standardize the data for CCA, we transformed the variables to ensure that they have comparable scales and to reduce the influence of outliers using the logarithmic transformation.

Finally, we used the Kendall rank correlation coefficient to assess the correlation between environmental variables and species abundance [57].

**3. Results**

Based on our registration of 45 phytosociological relevés, we have identified eight dominant plant communities in the area. These communities are primarily dominated by *N. stricta*, *Festuca violacea*, *K. myosuroides*, *F. amethystina*, *F. airoides*, *S. rigida*, *F. versicolor* and *F. carpatica*. In total, we have identified 235 plant species from 40 different families. Among these, 30 species are endemic, rare, or vulnerable to the studied territory, highlighting the importance of conserving this region's biodiversity (Figure 3). We also calculated the abundance of species (Table 2).

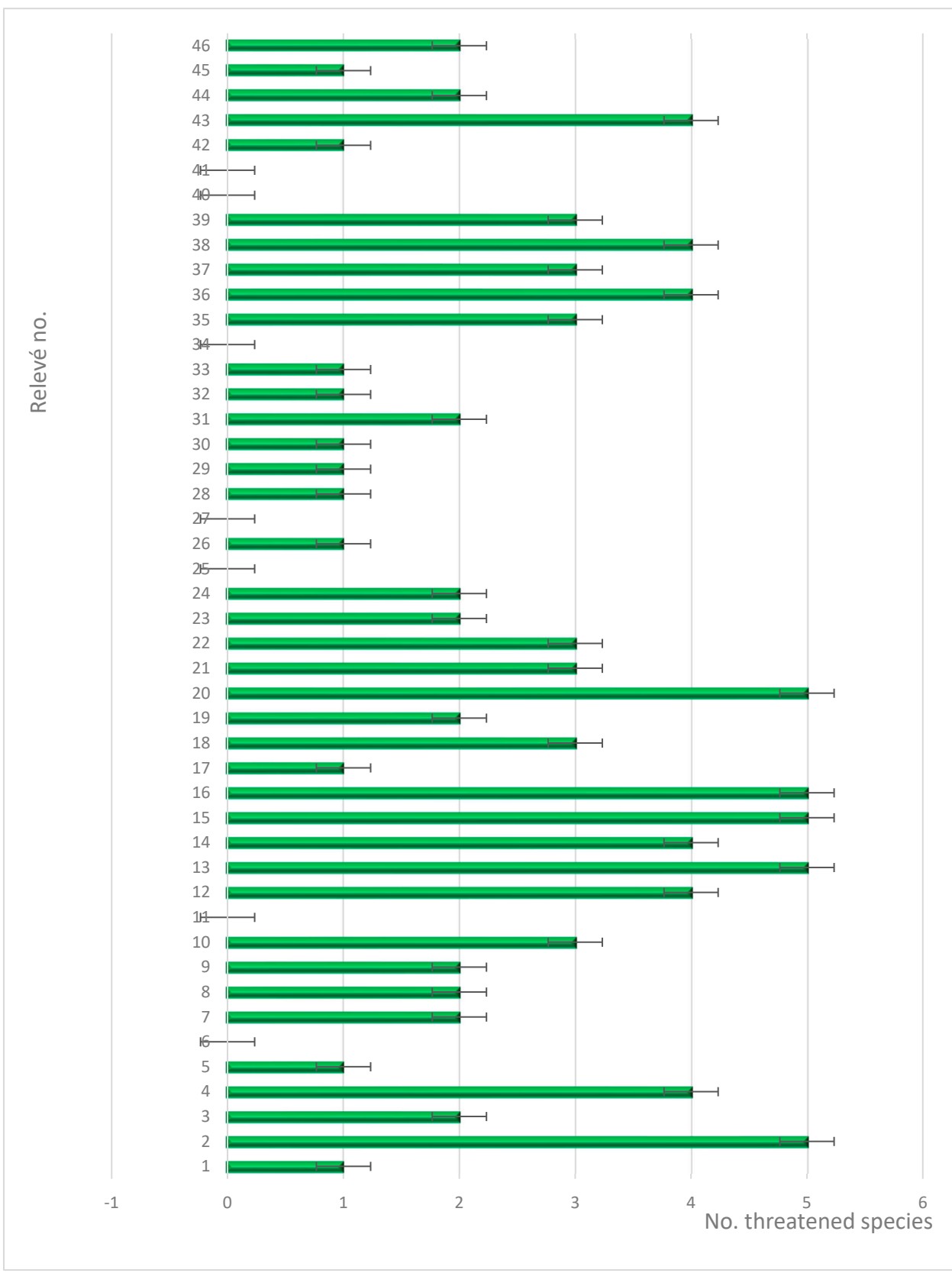

**Figure 3.** Abundance of threatened species in relevés.

**Table 2.** Ecological metrics by taxa.

| No. | 1 | 2 | 3 | 4 | 5 | 6 | 7 | 8 | 9 | 10 | 11 | |
|---|---|---|---|---|---|---|---|---|---|---|---|---|
| Taxa_S | 26 | 22 | 31 | 22 | 15 | 7 | 5 | 13 | 16 | 21 | 25 | |
| Individuals | 3 | 6 | 5 | 7 | 8 | 2 | 3 | 6 | 5 | 5 | 8 | |
| Dominance_D | 0.0489 | 0.0947 | 0.0638 | 0.1166 | 0.1358 | 0.22 | 0.4 | 0.1694 | 0.1078 | 0.0791 | 0.0612 | |
| Simpson_1-D | 0.9511 | 0.9053 | 0.9362 | 0.8834 | 0.8642 | 0.78 | 0.6 | 0.8306 | 0.8922 | 0.9209 | 0.9388 | |
| Shannon_H | 3.17 | 2.808 | 3.201 | 2.715 | 2.372 | 1.748 | 1.228 | 2.209 | 2.548 | 2.849 | 3.05 | |
| Evenness_eˆH/S | 0.9158 | 0.7534 | 0.7925 | 0.6864 | 0.7143 | 0.8205 | 0.6826 | 0.7004 | 0.7985 | 0.8226 | 0.8447 | |
| Brillouin | 0.2665 | 0.3675 | 0.2622 | 0.3656 | 0.5972 | 0.1449 | 0.0966 | 0.419 | 0.397 | 0.3693 | 0.6347 | |
| Menhinick | 6.713 | 5.588 | 7.02 | 5.416 | 4.082 | 3.13 | 2.236 | 3.92 | 4.718 | 5.612 | 5.976 | |
| Margalef | 22.76 | 11.72 | 18.64 | 10.79 | 6.733 | 8.656 | 3.641 | 6.697 | 9.32 | 12.43 | 11.54 | |
| Equitability_J | 0.973 | 0.9084 | 0.9323 | 0.8783 | 0.8758 | 0.8983 | 0.7627 | 0.8612 | 0.9188 | 0.9359 | 0.9476 | |
| Fisher_alpha | 0 | 0 | 0 | 0 | 0 | 0 | 0 | 0 | 0 | 0 | 0 | |
| Berger-Parker | 0.1333 | 0.2581 | 0.2051 | 0.303 | 0.2963 | 0.4 | 0.6 | 0.3636 | 0.2609 | 0.2143 | 0.1714 | |
| Chao-1 | 26 | 23 | 31 | 23 | 15.5 | 7 | 5 | 14 | 17 | 22 | 35 | |
| **No.** | **12** | **13** | **14** | **15** | **16** | **17** | **18** | **19** | **20** | **21** | **22** | |
| Taxa_S | 24 | 16 | 14 | 22 | 20 | 21 | 25 | 26 | 14 | 22 | 22 | |
| Individuals | 7 | 7 | 6 | 9 | 11 | 8 | 8 | 10 | 5 | 3 | 3 | |
| Dominance_D | 0.083 | 0.1302 | 0.1198 | 0.089 | 0.1015 | 0.0781 | 0.0648 | 0.0744 | 0.1653 | 0.0782 | 0.0592 | |
| Simpson_1-D | 0.917 | 0.8698 | 0.8802 | 0.911 | 0.8985 | 0.9219 | 0.9352 | 0.9256 | 0.8347 | 0.9218 | 0.9408 | |
| Shannon_H | 2.915 | 2.458 | 2.413 | 2.803 | 2.646 | 2.827 | 3.015 | 2.97 | 2.272 | 2.898 | 2.992 | |
| Evenness_eˆH/S | 0.7685 | 0.7303 | 0.7979 | 0.7496 | 0.7048 | 0.8041 | 0.8158 | 0.7495 | 0.6927 | 0.8242 | 0.9053 | |
| Brillouin | 0.4566 | 0.5228 | 0.545 | 0.6277 | 0.7379 | 0.6283 | 0.5853 | 0.6379 | 0.2781 | 0.1879 | 0.2703 | |
| Menhinick | 5.821 | 4.438 | 4.221 | 5.259 | 4.65 | 5.25 | 5.893 | 5.742 | 4.221 | 5.988 | 6.102 | |
| Margalef | 11.82 | 7.708 | 7.255 | 9.558 | 7.924 | 9.618 | 11.54 | 10.86 | 8.077 | 19.12 | 19.12 | |
| Equitability_J | 0.9172 | 0.8866 | 0.9145 | 0.9067 | 0.8832 | 0.9284 | 0.9368 | 0.9115 | 0.8609 | 0.9374 | 0.9678 | |
| Fisher_alpha | 0 | 0 | 0 | 0 | 0 | 0 | 0 | 0 | 0 | 0 | 0 | |
| Berger-Parker | 0.2353 | 0.3077 | 0.2727 | 0.2286 | 0.2162 | 0.1875 | 0.1667 | 0.1951 | 0.3636 | 0.2222 | 0.1538 | |
| Chao-1 | 27 | 19 | 17 | 23.5 | 20.5 | 22.5 | 26.5 | 26.33 | 14 | 22 | 22 | |
| **No.** | **23** | **24** | **25** | **26** | **27** | **28** | **29** | **30** | **31** | **32** | **33** | |
| Taxa_S | 22 | 23 | 16 | 14 | 26 | 11 | 19 | 11 | 35 | 25 | 29 | |
| Individuals | 5 | 2 | 6 | 7 | 8 | 8 | 8 | 6 | 8 | 13 | 9 | |
| Dominance_D | 0.0749 | 0.0562 | 0.136 | 0.1215 | 0.0525 | 0.1493 | 0.1094 | 0.2018 | 0.042 | 0.0734 | 0.0816 | |
| Simpson_1-D | 0.9251 | 0.9438 | 0.864 | 0.8785 | 0.9475 | 0.8507 | 0.8906 | 0.7982 | 0.958 | 0.9267 | 0.9184 | |
| Shannon_H | 2.901 | 3.045 | 2.443 | 2.384 | 3.121 | 2.126 | 2.599 | 1.988 | 3.414 | 2.925 | 2.997 | |
| Evenness_eˆH/S | 0.8269 | 0.9133 | 0.7189 | 0.7745 | 0.8723 | 0.7616 | 0.7081 | 0.6639 | 0.868 | 0.7455 | 0.6902 | |
| Brillouin | 0.3649 | 0.2044 | 0.3977 | 0.604 | 0.6463 | 0.7193 | 0.4983 | 0.3614 | 0.5474 | 0.8796 | 0.4357 | |
| Menhinick | 5.777 | 6.379 | 4.525 | 4.041 | 6.128 | 3.244 | 4.75 | 3.395 | 7.379 | 5.33 | 6.183 | |
| Margalef | 13.05 | 31.74 | 8.372 | 6.681 | 12.02 | 4.809 | 8.656 | 5.581 | 16.35 | 9.357 | 12.74 | |
| Equitability_J | 0.9385 | 0.9711 | 0.881 | 0.9032 | 0.9581 | 0.8865 | 0.8828 | 0.8292 | 0.9602 | 0.9088 | 0.8899 | |
| Fisher_alpha | 0 | 0 | 0 | 0 | 0 | 124.6 | 0 | 0 | 0 | 0 | 0 | |
| Berger-Parker | 0.2069 | 0.1538 | 0.32 | 0.25 | 0.1111 | 0.2609 | 0.25 | 0.381 | 0.1333 | 0.1364 | 0.1818 | |
| Chao-1 | 23 | 23 | 17 | 14.5 | 28 | 11 | 19 | 11 | 45 | 31 | 29 | |
| **No.** | **34** | **35** | **36** | **37** | **38** | **39** | **40** | **41** | **42** | **43** | **44** | **45** |
| Taxa_S | 38 | 22 | 18 | 20 | 16 | 18 | 10 | 25 | 15 | 13 | 12 | 11 |
| Individuals | 15 | 10 | 5 | 8 | 8 | 7 | 6 | 8 | 6 | 4 | 5 | 4 |
| Dominance_D | 0.0486 | 0.0825 | 0.1243 | 0.0822 | 0.1029 | 0.1177 | 0.22 | 0.0679 | 0.1458 | 0.1191 | 0.1111 | 0.1696 |
| Simpson_1-D | 0.9515 | 0.9175 | 0.8757 | 0.9178 | 0.8971 | 0.8823 | 0.78 | 0.9321 | 0.8542 | 0.8809 | 0.8889 | 0.8304 |
| Shannon_H | 3.378 | 2.805 | 2.565 | 2.774 | 2.538 | 2.555 | 1.887 | 2.983 | 2.369 | 2.361 | 2.351 | 2.119 |
| Evenness_eˆH/S | 0.7717 | 0.7512 | 0.7222 | 0.8013 | 0.7911 | 0.7148 | 0.6598 | 0.79 | 0.7127 | 0.8153 | 0.8749 | 0.7568 |
| Brillouin | 0.8725 | 0.6963 | 0.2725 | 0.6407 | 0.6999 | 0.4459 | 0.3674 | 0.5385 | 0.4042 | 0.3285 | 0.5623 | 0.291 |
| Menhinick | 6.996 | 5.115 | 4.992 | 5.08 | 4.355 | 4.727 | 3.162 | 5.812 | 4.33 | 4.218 | 4 | 3.773 |
| Margalef | 13.66 | 9.12 | 10.56 | 9.137 | 7.213 | 8.736 | 5.023 | 11.54 | 7.814 | 8.656 | 6.835 | 7.213 |
| Equitability_J | 0.9287 | 0.9075 | 0.8874 | 0.926 | 0.9155 | 0.8838 | 0.8194 | 0.9268 | 0.8749 | 0.9204 | 0.9462 | 0.8838 |
| Fisher_alpha | 0 | 0 | 0 | 0 | 0 | 0 | 0 | 0 | 0 | 0 | 0 | 0 |
| Berger–Parker | 0.1356 | 0.1622 | 0.3077 | 0.1935 | 0.2222 | 0.2759 | 0.4 | 0.1622 | 0.3333 | 0.2105 | 0.2222 | 0.3529 |
| Chao-1 | 45.5 | 22.5 | 18 | 21.5 | 17.5 | 18 | 10 | 25 | 16 | 13 | 13.5 | 11 |

Our analysis shows that around half of the identified plant species belong to four main families, which are *Asteraceae* (52 species), *Poaceae* (37 species), *Caryophyllaceae* (18 species) and *Fabaceae* (11 species) (Figure 4).

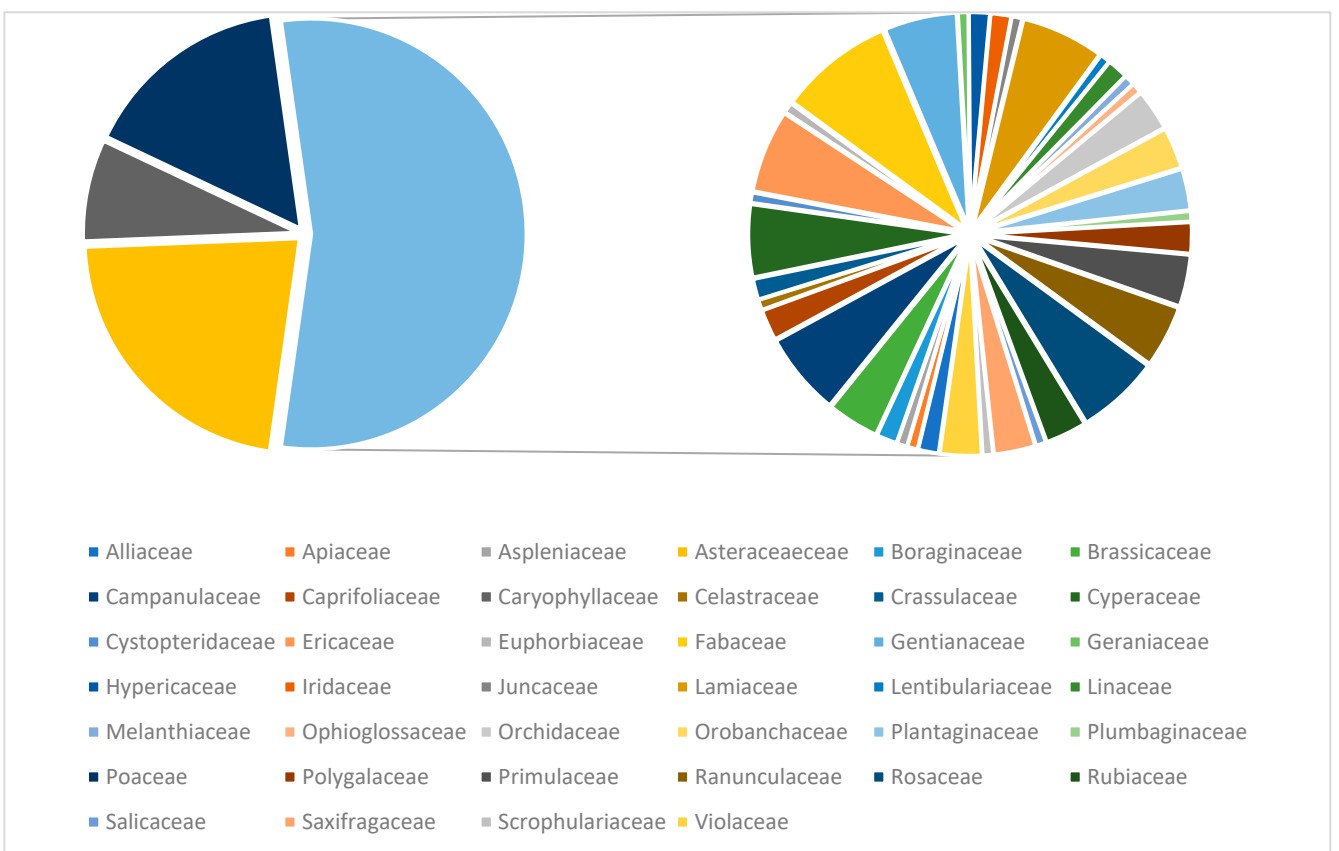

**Figure 4.** The families from studied area.

### 3.1. Species Diversity

According to Table 2, the CCA analysis revealed that relevés 1 and 3 are dominated by *S. rigida* and 31, 33 and 34 are dominated by *F. violacea* with a higher species count and thus a greater diversity (H' = 3.17, H' = 3.20, H' = 3.41, H' = 2.99 and H' = 3.37) compared to other relevés with lower species counts (6 and 7 dominated by *S. rigida*, 8 dominated by *Festuca amethystine* and 28 dominated by *F. airoides*). Conversely, the latter relevés had reduced diversity with H' values of 1.74, 1.22, 2.20 and 2.12. Moreover, the evenness values (E) for the dominant communities indicated a uniform distribution of species abundances, with higher values (E = 0.91, E = 0.79, E = 0.86, E = 0.74 and E = 0.77) for relevés 1, 3, 31, 33 and 34, while lower values (E = 0.82, E = 0.68, E = 0.70 and E = 0.76) for relevés 6, 7, 8 and 28 indicated an uneven distribution of species abundance. These summarized findings are presented in Table 1.

### 3.2. The Canonical Correspondence Analyses

Canonical correspondence analysis was performed to examine the relationships between floristic composition and environmental variables. The first canonical axis, with a higher eigenvalue ($\lambda_1$ = 0.45), represents the sampling area as the main gradient, followed by the second and third axes with lower eigenvalues ($\lambda_2$ = 0.25 and $\lambda_3$ = 0.11, respectively). Vegetation coverage is represented by the second axis. The total inertia was 4.03. The highest correlations between species and environmental variables were observed for the first (r = 0.81) and second (r = 0.81) axes, whereas the third axis showed the lowest correlation (r = 0.67). The cumulative percentage variance of species data was as follows: 11.4% for

the first axis, 17.7% for the second axis and 20.6% for the third axis. The variance of the species–environment relationship was 46.5% for the first axis, 72.6% for the second axis and 84.2% for the third axis. The Monte Carlo test of significance of the first canonical axis was an F-ratio = 4.999 ($p < 0.01$) and for all canonical axes, the trace was 0.98 and F-ratio was 2.521 ($p < 0.01$).

Please note that the results of the canonical correspondence analysis (CCA) should be interpreted with caution due to the potential instability caused by the ratio of variables (species plus environmental variables) to relevés. It is recommended to have a minimum of three times the number of relevés compared to the number of variables for eigen analysis-based multivariate analysis. Given the specific characteristics of our dataset, where the number of variables exceeds the number of relevés, the stability of the CCA results may be compromised.

*Agrostis capillaris* subsp. *capillaris* and *Veronica officinalis* showed the highest positive correlation with axis 1, which is dependent on the sampling area gradient. *Agrostis rupestris*, *F. airoides* and *Luzula sudetica* were highly correlated with axis 2, with their abundance increasing with vegetation coverage. *Dryas octopetala*, on the other hand, showed a negative correlation with axis 1 (Figure 5).

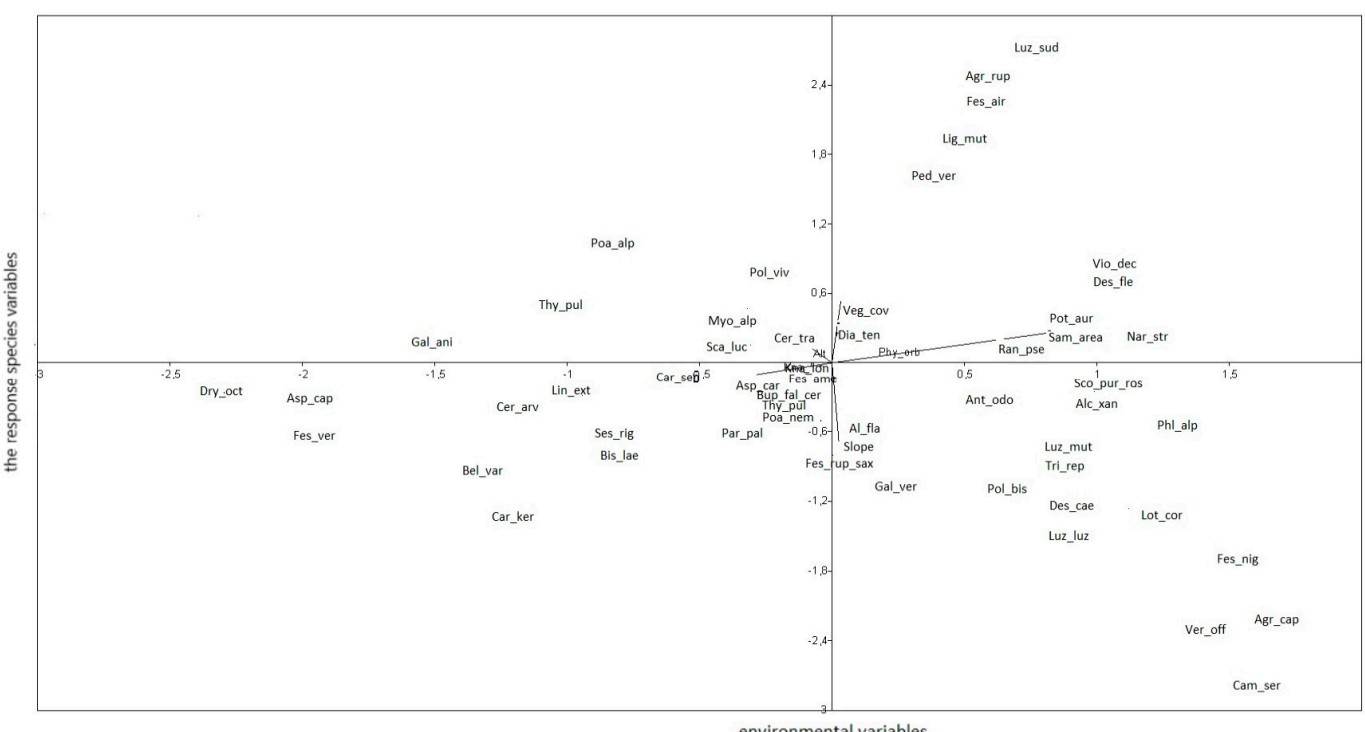

**Figure 5.** Relationships between abundances of species and environmental variables.

Figure 5 shows that the abundances of *Agrostis capillaris* subsp. *capillaris*, *Deschampsia caespitosa*, *Festuca nigrescens*, *N. stricta*, *Polygonum bistorta*, *Potentilla aurea* subsp. *chrysocraspeda*, *Trifolium repens* and *Veronica officinalis* decrease with increasing altitude. Conversely, the abundances of *Asperula capitata*, *Bellardiochloa variegata*, *Carex sempervirens*, *Cerastium arvense* subsp. *lerchenfeldianum*, *Dianthus tenuifolius*, *F. amethystina*, *F. versicolor*, *Linum perenne* subsp. *extraaxillare*, *Polygonum viviparum*, *Scabiosa lucida* and *Thymus pulegioides* increase with altitude.

Regarding slope, the abundances of *Agrostis capillaris* subsp. *capillaris*, *Carex sempervirens*, *Deschampsia caespitosa*, *Festuca nigrescens*, *Lotus corniculatus*, *Luzula luzuloides* subsp. *cuprina*, *Parnassia palustris* and *Veronica officinalis* increase with slope, whereas the abundances of *Agrostis rupestris*, *Ligusticum mutellina* and *Pediculris verticillata* decrease with slope.

Finally, *Agrostis rupestris*, *F. airoides*, *Ligusticum mutellina* and *Luzula sudetica* show a positive correlation with vegetation coverage, while the abundances of *Agrostis capillaris* subsp. *capillaris*, *Carex sempervirens*, *Deschampsia caespitosa*, *Dryas octopetala*, *Festuca rupicola* subsp. *saxatilis*, *Lotus corniculatis*, *Luzula multiflora* and *Veronica officinalis* decrease with increasing vegetation coverage.

The result of the Pearson correlation coefficient (r = 0.170) indicates a very weak correlation between the plot area and the number of species in a plot (Table 3). This suggests that there is no significant linear relationship between the measured area and the number of species observed in a given area.

**Table 3.** Summary of the CCA performed on correlations between response species variables and environmental variables on Kendall's rank correlation coefficients (significant results in bold).

| Variable | Axis 1 | Axis 2 | Axis 3 |
|---|---|---|---|
| Alt | **0.35 \*\*\*** | 0.14 | 0.07 |
| Asp | −0.15 | −0.10 | **−0.23 \*** |
| Slope | 0.02 | −0.61 | **−0.21 \*** |
| Veg_cov | 0.11 | **0.39 \*\*\*** | −0.17 |
| Agr_cap_cap | 0.47 | **0.35 \*\*\*** | **0.27 \*\*** |
| Agr_rup | 0.14 | 0.54 | 0.16 |
| Al_fla | 0.14 | 0.00 | −0.01 |
| Alc_xan | **0.22 \*** | 0.03 | **0.37 \*\*\*** |
| Ant_odo | **0.37 \*\*\*** | 0.07 | −0.18 |
| Asp_cap | −0.44 | −0.03 | 0.19 |
| Bel_var | **0.35 \*\*\*** | −0.04 | −0.18 |
| Bis_lae | **−0.22 \*** | −0.01 | **−0.21 \*** |
| Bup_fal_cer | **−0.27 \*\*** | −0.19 | **−0.33 \*\*** |
| Cam_ser | **0.31 \*\*** | **−0.30 \*\*** | **0.24 \*** |
| Car_ker | **−0.32 \*\*** | **−0.22 \*** | −0.02 |
| Car_sem | −0.44 | −0.10 | **−0.33 \*\*** |
| Cer_arv_ler | −0.43 | −0.01 | −0.12 |
| Cer_tra | −0.04 | −0.05 | **−0.28 \*\*** |
| Des_cae | **0.25 \*** | **−0.25 \*** | 0.02 |
| Des_fle | 0.42 | −0.00 | 0.14 |
| Dia_ten | −0.09 | −0.00 | **−0.31 \*\*** |
| Dry_oct | −0.47 | 0.11 | 0.43 |
| Fes_air | **0.25 \*** | 0.51 | **0.25 \*** |
| Fes_ame | −0.10 | 0.05 | −0.54 |
| Fes_nig | 0.59 | **−0.31 \*\*** | **0.34 \*\*\*** |
| Fes_rup_sax | −0.10 | −0.17 | 0.11 |
| Fes_ver | −0.51 | −0.06 | 0.03 |
| Gal_ani | **−0.33 \*\*** | 0.08 | 0.04 |
| Gal_ver | **0.23 \*** | **−0.28 \*\*** | 0.13 |
| Kna_lon | −0.14 | −0.07 | **−0.24 \*** |
| Lig_mut | **0.24 \*** | **0.31 \*\*** | 0.04 |

**Table 3.** *Cont.*

| Variable | Axis 1 | Axis 2 | Axis 3 |
|---|---|---|---|
| Lin_per_ext | −0.42 | −0.14 | −0.07 |
| Lot_cor | 0.15 | **−0.30 \*\*** | 0.13 |
| Luz_luz_cup | **0.22 \*** | −0.41 | 0.03 |
| Luz_mul | 0.42 | −0.06 | −0.03 |
| Luz_sud | **0.32 \*\*** | 0.40 | 0.02 |
| Myo_alp | **−0.25 \*** | −0.10 | −0.13 |
| Nar_str | 0.58 | −0.03 | **0.30 \*\*** |
| Par_pal | −0.07 | −0.01 | −0.40 |
| Ped_ver | **0.24 \*** | 0.15 | **0.25 \*** |
| Phl_alp | 0.40 | −0.06 | 0.15 |
| Phy_orb | −0.10 | −0.01 | **−0.29 \*\*** |
| Poa_alp | −0.17 | **0.21 \*** | **0.21 \*** |
| Poa_nem | −0.12 | −0.14 | **−0.27 \*\*** |
| Pol_bis | 0.13 | **−0.25 \*** | 0 |
| Pol_viv | −0.19 | **0.28 \*\*** | −0.07 |
| Pot_aur_chr | 0.57 | **0.21 \*** | 0.08 |
| Ran_pse | 0.42 | −0.06 | 0.12 |
| Sca_luc | **−0.26 \*** | 0.00 | −0.16 |
| Sco_pur_ros | 0.48 | −0.06 | 0.17 |
| Ses_rig_hay | −0.49 | **−0.33 \*\*** | −0.18 |
| Thy_pul | −0.17 | 0.17 | **0.26 \*** |
| Thy_pur | −0.08 | 0.06 | −0.46 |
| Tri_rep | **0.36 \*\*\*** | **−0.21 \*** | **0.35 \*\*\*** |
| Ver_off | 0.47 | −0.40 | **0.32 \*\*** |
| Vio_dec | **0.38 \*\*\*** | 0.13 | 0.11 |

Significance levels for correlation coefficients are indicated as follows: * ($p < 0.05$), ** ($p < 0.01$), and *** ($p < 0.001$).

The results of the statistical analysis for alpine grassland variables are presented in Table S1. This legend briefly describes the contents of the table and specifies the variables analyzed. It gives an overview of the results of the statistical analysis, including test statistics such as Shapiro–Wilk W, *p*-values (normal and Monte Carlo), Jarque–Bera JB and Chi-square values. In addition, it highlights the result of the "OK" Chi-square test for each variable based on the criterion of having a sample size (N) greater than 20.

Based on the information provided, the canonical correspondence analysis was used to understand the relationships between floristic composition and environmental variables. The analysis revealed three canonical axes, with the first axis being the most significant, representing the sampling area as the main gradient. The second axis is represented by vegetation coverage, while the third axis has a lower correlation between species and environmental factors. The canonical correspondence analysis revealed significant relationships between floristic composition and environmental variables, with the sampling area and vegetation coverage being the main factors. The analysis highlights the species that are most strongly correlated with each axis and provides valuable insights into the factors influencing the distribution of plant species in the area.

Figure 6 provides information on the elevations of each studied community. In general, grasslands above 1500 m are mainly used for grazing sheep and cattle, and are considered

high-mountain pastures [5]. We also compiled a table with notes on threatened species [59] to highlight their presence once again. We also added the conservation value [60] of these grasslands.

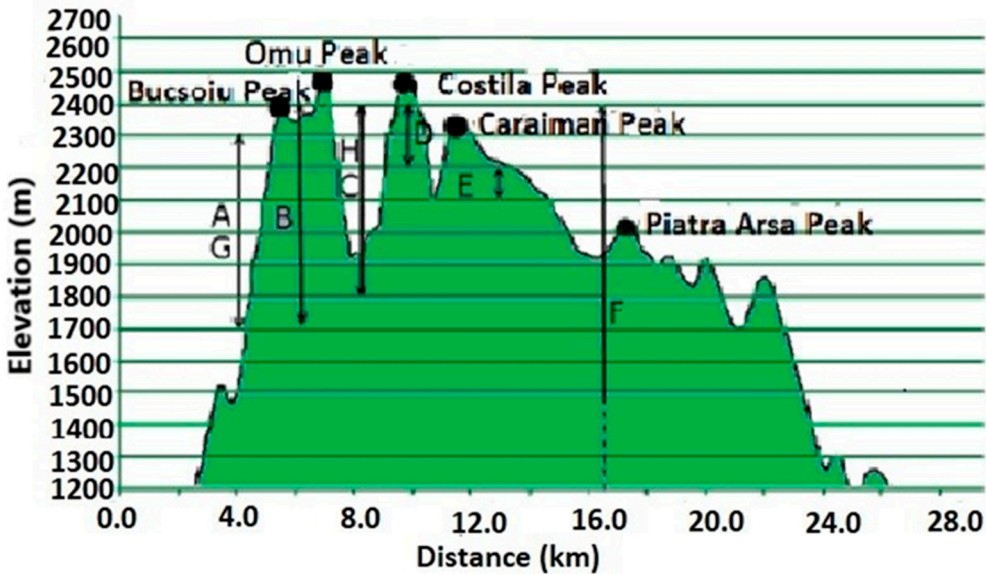

**Figure 6.** Communities dominated by A: *N. stricta* (45.42° N, 25.48° E), B: *F. amethystina* (45.35° N, 25.48° E), C: *F. airoides* (45.35° N, 25.48° E), D: *K. myosuroides* (45.38° N, 25.48° E), E: *F. versicolor* (45.41° N, 25.49° E), F: *S. rigida* (45.43° N, 25.44° E), G: *F.violacea* (45.38° N, 25.48° E) and H: *F. carpatica* (45.44° N, 25.45° E).

Moreover, we add a table that provides a comprehensive overview of the conservation status of various communities based on EUNIS2020 [61,62], Red List [63] and NATURA 2000 codes [64]. It includes information on threatened species within each habitat, as well as their corresponding conservative value [61,65] (Table 4).

**Table 4.** Studied Communities Conservation Status.

| Community | EUNIS2020 Classification | Red List Category | NATURA 2000 Habitat Code | Threatened Species | Conservative Value |
|---|---|---|---|---|---|
| *K. myosuroides* | N11/Alpine and sub-Alpine grassland | E4.4a Arctic-alpine calcareous grassland | 6170 Alpine and subalpine calcareous grasslands | 9: *Achillea oxyloba* subsp. *schurii, Oxytropis carpatica, Poa molinerii* subsp. *glacialis, Thymus pulcherrimus, Astragalus alpinus, Chamorchis alpina, Erigeron uniflorus, K. myosuroides, Viola alpina* | High, Endemic habitat |
| *F. airoidis* | N11/Alpine and sub-Alpine grassland | E4.3b Temperate acidophilous alpine grassland | 6150 Siliceous alpine and boreal grasslands | 7: *Cerastium arvense* subsp. *lerchenfeldianum, Dianthus glacialis* subsp. *gelidus, Rhododendron myrtifolium, Thymus pulcherrimus, Loiseleuria procumbens, Lomatogonium carinthiacum, Saxifraga oppositifolia* | High, Endemic habitat |

Table 4. *Cont.*

| Community | EUNIS2020 Classification | Red List Category | NATURA 2000 Habitat Code | Threatened Species | Conservative Value |
|---|---|---|---|---|---|
| *F. versicolor* | N11/Alpine and sub-Alpine grassland | E4.4a Arctic-alpine calcareous grassland | 6170 Alpine and subalpine calcareous grasslands | 8: *Achillea oxyloba* subsp. *schurii, Anthemis carpatica, Cerastium arvense* subsp. *lerchenfeldianum, Dianthus glacialis* subsp. *gelidus, Dianthus spiculifolius, Linum perenne* subsp. *extraaxillare, Onobrychis transsilvanica, Thymus pulcherrimus* | High, Endemic habitat |
| *F. amethystina:* | N11/Alpine and sub-Alpine grassland | E4.4a Arctic-alpine calcareous grassland | 6170 Alpine and subalpine calcareous grasslands | 10: *Cerastium arvense* subsp. *lerchenfeldianum, Coeloglossum viride, Dianthus glacialis* subsp. *gelidus, Dianthus spiculifolius, Linum perenne* subsp. *extraaxilare, Leontopodium alpinum, Lomatogonium carthiacum, Onobrychis transsilvanica, Nigritella rubra, Thymus pulcherrimus* | High, Endemic habitat |
| *F. violacea* | N11/Alpine and sub-Alpine grassland | E4.3b Temperate acidophilous alpine grassland | 6230* Species-rich Nardus grasslands, on siliceous substrates in mountain areas | 4: *Bruckenthalia spiculifolia, Campanula patula* subsp. *abietina, Cerastium fontanum* subsp. *macrocarpum, Coeloglossum viride, Pinguicula vulgaris* | Moderate, Endemic habitat, European priority |
| *N. stricta* | N11/Alpine and sub-Alpine grassland | E4.3b Temperate acidophilous alpine grassland | 6230* Species-rich Nardus grasslands, on siliceous substrates in mountain areas | 4: *Campanula patula* subsp. *abietina, Dianthus glacialis* subsp. *gelidus, Pseudorchis albida, Thymus pulcherrimus* | Moderate, European priority |
| *S. haynaldiana* | N11/Alpine and sub-Alpine grassland | E4.4a Arctic-alpine calcareous grassland | 6170 Alpine and subalpine calcareous grasslands | 8: *Centaurea pinnatifida, Cerastium arvense* subsp. *lerchenfeldianum, C. transsilvanicum, Dianthus spiculifolius, Gentiana lutea, Linum perenne* subsp. *extraaxillare, Onobrychis transsilvanica* and *Leontopodium alpinum* | High, Endemic habitat |
| *F. carpaticae* | N11/Alpine and sub-Alpine grassland | E4.4a Arctic-alpine calcareous grassland | 6170 Alpine and subalpine calcareous grasslands | 9: *Achillea oxyloba* subsp. *schurii, Carduus kerneri* subsp. *kerneri, Cerastium transsilvanicum, Doronicum carpaticum, F. carpatica, Leucanthemum waldsteini, Ligularia sibirica, Linum perenne* subsp. *extraaxillare, Sesleria bielzii* | High, Endemic habitat |

## 4. Discussion

### 4.1. Ecology of Studied Communities

In the Bucegi Massif of the South-Eastern Carpathians, alpine grasslands cover a wide altitudinal range, extending from the upper mountain belt around 1600 m above sea level to the alpine belt at 2400 m [43,44,49]. The dominant graminoid community in the lower alpine zone, where siliceous rocks prevail [45], is *N. stricta*, while *F. rubra* and *S. rigida* communities thrive on calcareous rocks. As we ascend to the upper alpine zone, *K. myosuroides*

and *F. violacea* communities tend to dominate [4,31]. These alpine grasslands encompass diverse plant communities such as oligo-mesotrophic subalpine grasslands (*Potentillo-Nardion*) and calcareous alpine grasslands (*Festuco-Seslerion bielzii*) [35,38]. *N. stricta*. We noticed in the Western Carpathians [62], alpine grassland communities are classified into *Caricetalia curvulae* and *Seslerietalia variae* orders, depending on the bedrock composition. *Caricetalia curvulae* communities prevail in areas with crystalline bedrock, while *Seslerietalia variae* communities are more abundant in regions with carbonate bedrock [38]. However, grasslands on acidic and nutrient-poor soils associated with crystalline bedrock display lower diversity but exhibit a remarkably similar composition [62].

We firstly refer to the environmental conditions that might differ between the different communities studied. These include factors such as altitude, temperature, rainfall, soil type, and exposure to sunlight [5,7,63,66]. How do these factors vary in regions where *N. stricta*, *F. rubra*, *S. rigida*, *K. myosuroides* and *F. violacea* communities predominate? Are there significant correlations between certain environmental parameters and the distribution of these plant communities? We found a correlation between the abundance of different plant species in each of the studied communities and altitude, with some species increasing in abundance as altitude increases, while others decrease [67,68]. Heegaard [69] proposed that changes in species abundance are influenced by both environmental factors related to temperature as well as biological interactions. Meanwhile, decreases in values of occurrence are likely due to the increased severity of the environment with increasing altitude, which is associated with temperature decreases [70]. There is also a correlation between the abundance of certain plant species and slope, with some species increasing in abundance with slope, while others decrease. Our results are supported by other studies assessing a strong effect of slope aspect on grassland productivity and species composition, in addition to other environmental factors such as water and soil temperature [66,71]. A similar study from the Po plain, karstic and pre-Alpine mountain regions and the western part of the Pannonian plain [17] indicates that the group of soil factors had correlated with both species richness and composition, followed by climatic and topographic factors. Altitude, soil pH, geographical gradient, frequency of flooding, mean annual temperature, date of mowing, humidity, annual precipitation and soil nutrient content were found to be the most important factors in explaining the variance of plant species composition [12,61,63]. Within a community, the floristic diversity is increasing as a function of the number of species and equitability of species abundance [63–65,72]. The results show that there is a dependence between species number and evenness in the assessment of floristic diversity. A greater number of species and an equitable distribution of abundance of species contribute to the increasing floristic diversity. According to the coefficient of variation, there is no greater variability of the diversity and evenness, but regarding the number of species within each relevé, it a greater heterogeneity of this variable was recorded.

Vegetation coverage also appears to be a significant factor in determining the abundance of different plant species [14,73], with some species positively correlated with vegetation coverage, while others are negatively correlated. Thus, there are studies that suggest that abundance is affected by species richness and by spatial scale [64]. Anyway, environmental heterogeneity is more important than the area in some studies [74,75].

The CCA analyses reveal a significant influence of the main species gradients. They present changes in plant diversity along the environmental gradients. Great importance is attributed to the distribution pattern along the altitude gradient, for instance, Trifolium repens is more abundant at low altitudes whereas Asperula capitata is more abundant at high altitudes. The slope and vegetation coverage are also gradients that indicate changes within the structure of studied assemblages [76,77]. The result is in accordance with the autecology of the investigated plants [78]. The plants significantly correlated to high altitudes prefer low temperatures and they are growing especially in the Alpine belt, whereas those of low altitudes prefer high temperatures [72,74].

The sampling area as the main gradient described by axis 1 is highly correlated to this axis. Also, the vegetation cover is highly correlated to axis 2. The slope influences, in a

positive way, the species abundance, probably due to the restricted accessibility of human activities.

*F. violacea*, *K. myosuroides*. These grasslands can either have a secondary origin, resulting from human activities such as grazing, logging or burning, or a primary origin, developing naturally over time. This information is supported by the research of Coldea and Cristea in 1998 [33], and Kyyak in 2004 [15]. We recorded a total of 235 plant species from 40 families, including 30 threatened species. This is consistent with previous studies that suggest that the highest concentration of threatened species can be found at high altitudes in the subalpine and lower alpine altitude belts [33]. Environmental factors, such as geomorphological and climatic variables, have played a significant role in fostering this richness of endemic, rare or vulnerable species in alpine regions [58].

The data suggest that the environment is moderately diverse, with 246 taxa present and a relatively even distribution of individuals among them.

Thus, relevés 1 and 3 (dominated by *S. rigida*) and 31, 33 and 34 (dominated by *F. violacea*) have the highest diversity but, at the same time, an equitable distribution of species abundance [59].

In contrast, relevés with a lower number of species (6 and 7 dominated by *S. rigida*, 8 dominated by *F. amethystina and* 28 dominated by *F. airoides*) have lower diversity and an uneven distribution of species abundance [59].

Analyzing the distribution of threatened species in the studied alpine grassland communities, we note an inconsistent distribution with large deviations. We try to explain this as being due to the local climate in alpine grasslands which can vary significantly over short distances, depending on factors such as exposure to wind and sun. This can create microhabitats that favor certain plant species over others [65].

Many alpine grasslands are grazed by herbivores such as cattle, sheep and goats [26,31]. From the point of view of interactions between dominant plant species and other organisms in their respective habitats, we can state, according to the literature [26,31], that there are herbivores in the studied area that preferentially feed on certain plant species, affecting their abundance and distribution [13,17]; this can affect plant distribution by altering resource availability and creating disturbances that favor certain plant species over others.

Our study of 46 alpine grassland communities has revealed a high number of threatened plant species. This is consistent with previous studies that suggest that the highest concentration of threatened plant species can be found at high altitudes in the subalpine and lower alpine altitude belt [5,10,12]. Environmental factors, such as geomorphological and climatic variables, have played a significant role in fostering this richness of endemic, rare and vulnerable species in alpine regions [65].

Comparing similar regions in southern Europe, it appears that mountain isolation has been more conducive to endemism than insularity [66]. Additionally, the degree of endemism tends to decrease as the overall number of plant species increases [67]. Rocky slopes, screes and alpine grasslands have been found to have the highest percentage of threatened plant species, consistent with the general distributional pattern of endemism in high-altitude Eurasian mountains [67].

### 4.2. Conservation of Studied Communities

The findings emphasize the importance of preserving the region's biodiversity. Studies in the high altitudes of the Iranian mountains have highlighted that they are an important hotspot for endemic species [68]. We also support the idea of continued botanical exploration of the mountain ranges, as well as taxonomic reviews of endemism-rich genera that have been understudied [68]. Given that our study area was contained within areas declared as protected, we consider it necessary to highlight habitat framing, even more so because it is recognized that the loss of habitat has been stopped within protected areas, but not in areas beyond their boundaries [70,71]. The EUNIS habitat classification system [72–74] is an important resource relied upon by professionals at various stages of nature conservation, including protected area design, resource inventories, monitoring,

management planning, impact assessments and ecological restoration target setting [75,76]. In this regard, we have included the EUNIS codes for each community studied and the European Red List of Habitats [56].

We studied different grassland communities in the Alpine belt that correspond to Natura 2000 codes for habitats [55]. Specifically, we found the following.

Communities of *F. violacea* and of *N. stricta* correspond to the 6230* species-rich *Nardus* grasslands in siliceous substrates in the mountain area (priority habitat) code. According to Sarbu et al. [33], they are the most common grasslands in the Bucegi Mountains and they are dominated by *N. stricta* and *F. airoides*, with 80% of the species requiring low temperatures and 70% needing moist conditions. In this habitat, we found 15 protected plant species, including rare, endemic and globally protected species that depend on low temperatures and prefer humid soil [77].

Communities of *F. airoides* that correspond to the 6150 Siliceous alpine and boreal grassland code. These short grasslands are considered a glacial relict and are found on the high crests of the mountains at alpine level [33], and they are dominated by species like *Agrostis rupestris*, *F. airoides* and *Potentilla aurea* subsp. *chrysocraspeda*. These communities thrive in low-temperature and humid conditions. The majority of the plants in these grasslands require low temperatures and moist soil for growth and development.

Communities of *K. myosuroides*, *F. versicolor*, *F. amethystina*, *S. rigida* and *F. carpatica* correspond to the 6170 Alpine and subalpine calcareous grassland code. They have a high biodiversity and are scientifically important for the South-Eastern Carpathians [78,79] where they are endemic [80].

Based on Sarbu et al. [33], grassland habitats 6150 and 6230* are considered to be very sensitive to warming and somewhat sensitive to drought. Moreover, another study revealed a decrease in the biodiversity on mountaintop calcareous alpine grasslands in the central Apennines [80].

In the opinion of García-González [81], Habitat 6170 does not require active management for conservation purposes due to its high structural complexity and fragility. Therefore, the most effective management approach is to refrain from intervening in it.

According to all studies, calcareous grasslands have a high conservation value, while grasslands in siliceous substrates have a moderate conservation value [82]. Moreover, most of them are endemic in the Carpathians: *K. myosuroides*, *F. versicolor*, *F. amethystina*, *S. rigida* and *F. carpatica* [82]. Additional field investigations will assess the current condition of these habitats, especially the priority Carpathian habitats [83].

A study of alpine habitats in the Eastern Styrian Alps has shown that climate change will have a significant and immediate impact on their flora and habitats. Fifteen species, including *Agrostis rupestris*, *Carex curvula*, *Primula minima*, *Dryas octopetala*, *Silene acaulis* and *Saxifraga paniculata*, which are also contained in the communities we studied, were identified as particularly vulnerable. By the end of the 21st century, the habitat suitability of these species will decrease significantly [84].

The alpine calcareous and siliceous grasslands of the Carpathian Mountains (identified by Natura 2000 code 6150, 6130, and 6170) are the habitat of a wide range of plant species of significant conservation value. These plants have specific environmental requirements, such as low temperatures, snow cover and high humidity, making them particularly vulnerable to the impacts of climate change. Plant species at risk include *Nigritella nigra*, a rare species, and *Dianthus glacialis* subsp. *gelidus*, an endemic species [33].

## 5. Conclusions and Future Prospects

The study highlights the influence of environmental factors, such as altitude, slope and vegetation cover, on the distribution and abundance of different plant communities in the alpine grasslands of the Bucegi Massif. Understanding these relationships is necessary for the conservation and effective management of these ecosystems.

The alpine grasslands in the study area display moderate floristic diversity, hosting a wide variety of plant species, including several threatened species. These grasslands

hold significant conservation value due to the presence of endemic, rare, and vulnerable plant species, making their preservation essential for maintaining biodiversity in the South-Eastern Carpathians. The observed distribution patterns of threatened species in the alpine grassland communities highlight the sensitivity of these ecosystems to even minor changes in environmental conditions [81–84]. The potential impacts of climate change on these habitats warrants further attention and necessitates long-term monitoring and conservation efforts [85–88].

The presence of high-conservation-value habitats, such as the Natura 2000-designated grassland communities [89], emphasizes the need for effective protection and management measures [84,85]. Proper management practices within protected areas [86], as well as in areas beyond their boundaries, are essential to halt habitat loss and ensure the preservation of these valuable ecosystems [89].

This study underscores the importance of continued botanical exploration and research in the mountain ranges, particularly in relation to taxonomic reviews of endemism-rich genera and threatened species. Additionally, ongoing monitoring of alpine grasslands will provide valuable insights into their responses to changing environmental conditions and help guide conservation efforts.

In conclusion, the study provides valuable insights into the distribution and composition of alpine grassland communities in the Bucegi Massif. The research highlights the importance of these ecosystems for biodiversity conservation and calls for strong conservation measures to protect them in the face of changing environmental conditions. Continued research and monitoring efforts are essential to ensure the long-term survival of these unique and ecologically significant habitats in the South-Eastern Carpathians.

The impact of herbivores on species richness is an important consideration for the conservation of protected areas. According to Fernández-Lugo et al. [2], grazing can have significant effects on vegetation patterns and community heterogeneity.

Grazing can also impact the structural diversity of plant communities. Reference [24] found that grazing had a significant effect on the structural heterogeneity of communities. Inappropriate grazing practices, such as overgrazing, can lead to soil erosion and vegetation changes [90–92], which in turn can negatively impact species richness and biodiversity [93].

An assessment of clues on how climate change or human activities affect the distribution and composition of these plant communities should be a priority for future research [94]. It is suspected that there has been a shift in the boundaries of some alpine areas and even an increase in the abundance of invasive species that may be influencing native plant communities [95].

**Supplementary Materials:** The following supporting information can be downloaded at https://www.mdpi.com/article/10.3390/su151612643/s1. Table S1: Data distribution.

**Author Contributions:** Conceptualization, C.B.-N.; methodology, C.B.-N. and F.Y.; software, C.B.-N. and F.Y.; validation, C.B.-N.; formal analysis, C.B.-N.; investigation, C.B.-N.; writing—original draft preparation, C.B.-N., F.Y. and O.K.; writing—review and editing, C.B.-N. and O.K.; visualization, C.B.-N. and O.K.; supervision, C.B.-N., F.Y. and O.K. All authors have read and agreed to the published version of the manuscript.

**Funding:** This research was funded by the project RO1567-IBB01/2022 of the Romanian Academy.

**Institutional Review Board Statement:** Not applicable.

**Informed Consent Statement:** Not applicable.

**Data Availability Statement:** The data are contained within the manuscript and Supplementary Materials.

**Acknowledgments:** The authors would like to express their appreciation to Ioana Vicol and Mihaela Ion and C.B.-N. would also like to extend sincere gratitude to Mario Pei for valuable advice.

**Conflicts of Interest:** The authors declare no conflict of interest.

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
