# Peer review of "Exploring the Biodiversity and Conservation Value of Alpine Grasslands in the Bucegi Massif, Romanian Carpathians"

_sustainability, doi:10.3390/su151612643_

Round 1

Reviewer 1 Report

Title:

ok.

Abstract:

·         Line 21: "Although the habitats have been grazing" - It seems like there might be a typographical error here. Did you mean "although the habitats have been grazed" or something else?

·          

Introduction

·         Line 56: Elaborate on the significant threats faced by grassland communities and their biota (e.g., "including habitat degradation, land-use changes, invasive species, and climate change").

·         Line 73: State the specific objective of the study

Materials and Methods

·         Provide the source/software used to create Figure 1.

·         Provide the source of data used to create Figure 2.

·         In line 95, it is stated that rainfall at altitudes of 1400-1800 m is about 1100 mm per year. However, it would be beneficial to provide more context or specify whether this is an average or a typical range for a better interpretation

·         In line 131, it is mentioned that the distribution of the data was assessed using the Shapiro-Wilk test. However, it would be helpful to provide more details on the results or implications of this analysis in relation to the study objectives.

·         Line 135 mentions "CANOCO version 4.5." It would be clearer to specify that CANOCO version 4.5 refers to a software package or program used for statistical analysis.

 Results:

1.      Reconsider Figure 3, and add a more accurate and clearer figure.

2.      Provide clearer Figure 5

3.      Provide clear Figure 6

Discussion

·         It would be beneficial to include specific references for the observations mentioned in the discussion.

·         Furthermore, discussing the ecological implications and potential reasons for the observed differences in community composition across the studied regions would be valuable. Are there any environmental factors or biotic interactions that could explain these patterns? Including a brief analysis or speculation on these aspects would greatly enrich the discussion.

·         Lastly, while the discussion covers the broad patterns of community composition, mentioning any limitations or gaps in the current understanding of these communities would be helpful. This could prompt future research directions and highlight areas that require further investigation.

Conclusions

·         Suggested changing, from conclusion to conclusion and future prospects

·         It would be beneficial if the authors could provide more specific recommendations for conservation and management strategies based on their findings. Suggestions for addressing the identified threats, such as inappropriate grazing, infrastructure construction, land use changes, and global warming, could be valuable for practitioners and policymakers.

·         Additionally, expanding on the effectiveness and efficiency of medium-term grazing exclusion for alpine meadows and steppes, as mentioned in the discussion section, would provide more practical insights for habitat restoration efforts

General comments

 After carefully reviewing the manuscript, I must commend the author for their skillful writing and overall presentation. However, I have identified several areas where the manuscript could be improved. These suggestions are intended to help the author further enhance the manuscript's readability, structure, and impact.

Careful proofreading is required. 

Author Response

Dear Reviewer,

Thank you for your comments concerning our manuscript. Those comments are all valuable and very helpful for revising and improving our paper, as well as the important guiding significance to our research. We have studied the comments carefully and have made corrections which we hope meet with approval. Revised portions are marked in the paper. 

Reviewer 2 Report

I have thoroughly read the manuscript article entitled “Exploring the Biodiversity and Conservation Value of Alpine 2 Grasslands in the Bucegi Massif, Romanian Carpathians” submitted in the Sustainability. Although article can be interesting for the readers of the Journal. Nevertheless, the language of the article is poor, along with long and ambiguous sentences. Proper arrangement of the references cited in the manuscript according to the Journal’s Guide Lines for the Authors. Pay attention to the use of comma i.e. punctuation. Check the length of the sentences and try to avoid long sentences until and unless it is needed.

 Abstract

Abstract always stands alone therefore pay attention on it. Abstract should be explicit, in proper flow with a strong and novel recommendation in the end. Avoid long sentences, and pay much attention on the main outcomes of the study. Readers should understand after reading the abstract that what actually had been done in this study. The abstract of the article in its current form is without any solid and scientific novelty, moreover, a lot of parenthesis are incorporated in the abstract. It should be improved and polished and only novel aspects should be highlighted and incorporated into the abstract.

Introduction

Introduction of the paper is too much important, a good introduction has to cover all the aspects of the studied problem. The current introduction of the article needs a lot of improvement. Sentences are not only really very long, but the actual studied problem or aspect is not covered in its true sense. Moreover, in some paragraphs, after the long sentence no reference is given? is this the authors on theory or notion? or aforementioned literature stated this? if aforementioned literature stated this, then where is the reference? Introduction is flow less, and without proper references, authors just tried to gather any information, not the concise and relevant information on the topic. Whole introduction should be revised carefully, with new, pertinent and relevant information on the studied topic.

Materials and Methods

This section of the article should be not too long, not too short, to the point and at the same time providing all the important details. After reading materials and methods section the reader can easily re-do or repeat it, and get the same results. The material and method section of the paper is very weak and readers can not re-do the work stated here. Statements are shabby and ambiguous, and sentences are long. Moreover, some important methods are just over-looked. They should be elaborated, in a proper sequence and in detail.

 Discussion

This section of the article is weak, authors mainly focused on their results but they did not discuss them according to international standards. Moreover the writing style of discussion section is also ambiguous, with long and weak sentences and in a repetitive way. I will recommend a thorough revision of discussion section.

Conclusion

Conclusion is not based on the actual results and findings of the authors, but based on the personal view points and notions of the authors. A strong and technically sound conclusion is highly recommended and needed in order to fulfil the intentional standards.

Language, wording and paraphrasing should be carefully reviewed and improved. A native English-speaking scientist or professional English editing service must edit your manuscript.

Author Response

Thank you for your comments concerning our manuscript. Those comments are all valuable and very helpful for revising and improving our paper, as well as the important guiding significance to our research. We have studied the comments carefully and have made corrections which we hope meet with approval. Hopefully, we have resolved all the issues you have raised in this review. Also, please let us know of any other changes that are required in this ms. This type of dialogue can foster further discussion and exchange of ideas, which is valuable in academia and research.

We are grateful for your thoughtful review. Your feedback is very valuable to me and I have made the necessary improvements to enhance the readability, structure and impact of the manuscript. Certainly, your contributions have helped me to refine the paper and make it even better.

Reviewer 3 Report

The entire manuscript needs extensive revision for language and grammar. Also fix the problems with alignment.

Introduction

Insufficient introduction. Elaborate and restructure it. Please write about the research gap and significance of the study.

  L 28-29- Add multiple references.

 L 40 – Remove ‘,’ after soil conservation. Please check for the unnecessary use of commas throughout the manuscript.

 L 42-56- The paragraph seems unnecessary. If you intended to highlight the importance of the study in policymaking, please rewrite it in a more concise manner.

L 57-63 – This paragraph here makes no sense to me. The authors made no mention of the study area's geographical location and then wrote about the Bucegi Mountains' sheep grazing tradition! Please remove it or rephrase and position it appropriately in the next paragraph.

 L 64-70- Inadequate review of previous research. Please specify the research gap.

 L 65 – Remove ‘also’ after ‘and’

 L 71-76- The objectives require more clarification.

 The generic part of a scientific name can be abbreviated after its first appearance in the manuscript. For example, Nardus stricta can be written as N. stricta later on after first use.

 Materials and methods

Information added below the figure 1 is insufficient. At least add the name of the study area since the authors didn’t label it on the map.  

Number the second subheading under material and methedology properly.

Write either fig or figure in the  footnotes of figures and make it uniform through out the manuscript.

Please mention  what the Y axis  mean in the fig 2, similarly fig 5 doesn't covey what the axes mean.

 Result

Make the subheading 3.2  in  uniform format and shift it to the next page.

 Make  table no. 4 more conveyable and use appropriate headings for each column.

 Discussion

Insufficient discussion. Need major revision.

Restructure the introduction to improve the aesthetic as well as logical flow

L 269-271- The same information is already given in the previous paragraph.

L 260-275- Merge both the paragraphs.

L 290-294- Irrelevant. Please remove it

 L 295-302- Remove and add it in the results section

 L 316- Add more references

 L 326-328- Add references

 L 349-351- What is the relevance of this statement here?

 L 355- A paragraph cannot be with a single sentence. Merge it to the next paragraph or add few more sentences.

 L 356- Missing ‘Period’ between the sentences.

  L 357-358- Paraphrase the sentence

 L 359-360 Check author guidelines for writing text reference. Avoid ‘in’ from the text references (Davies and Moss in 1998, Davies et al. in 2004, Moss in 2008). Check the entire manuscript for the same and correct accordingly.

 Conclusion

L 407-408- I couldn’t find any clear establishment of the relationship between the species abundance and environmental factors in the study. Might be due to the lack of proper discussion. So please consider rewriting it accordingly.

Could have included suggestions for further research

 Reference

Please go through the references, many are incomplete and make them all in a uniform format.

The entire manuscript needs extensive revision for language and grammar. Also fix the problems with alignment.

Author Response

We are grateful for your thoughtful review. Your feedback is very valuable to me and I have made the necessary improvements to enhance the readability, structure and impact of the manuscript. Certainly, your contributions have helped me to refine the paper and make it even better.

Round 2

Reviewer 1 Report

Thank you for accepting all the recommendations. But carefully proofread is required. 

Recheck the manuscript again for grammatical errors.